



# Horizontal electric fields from flow of auroral O$^+$($^2$P) ions at sub-second resolution

Sam Tuttle[1], Betty Lanchester[1], Björn Gustavsson[2], Daniel Whiter[1], Nickolay Ivchenko[3], Robert Fear[1], and Mark Lester[4]

[1]Physics and Astronomy, University of Southampton, Southampton, UK
[2]Department of Physics and Technology, University of Tromsø, Tromsø, Norway
[3]KTH, Royal Institute of Technology, Stockholm, Sweden
[4]Department of Physics and Astronomy, University of Leicester, Leicester, UK

**Correspondence:** B. S. Lanchester (B.S.Lanchester@soton.ac.uk)

**Abstract.** Electric fields are a ubiquitous feature of the ionosphere and are intimately linked with aurora through particle precipitation and field-aligned currents. We present a unique method to estimate ionospheric electric fields beside a dynamic auroral feature by solving the continuity equation for the metastable O$^+$($^2$P) ions, which emit as they move under the influence of electric fields during their 5 s lifetime. Simultaneous measurements of emission at 732.0 nm (from the O$^+$($^2$P) ions), and prompt emissions at 673.0 nm (N$_2$) and 777.4 nm (O), all at high spatial (100 m) and temporal (0.05 s) resolution, are used in the solution of the continuity equation, which gives the dynamic changes of the O$^+$ ion population at all heights in a 3D volume close to the magnetic zenith. Perspective effects are taken into account by a new geometric method, which is based on an accurate estimate of the magnetic zenith position. The emissions resulting from the metastable ions are converted to brightness images by projecting onto the plane of the ground, which are compared with the measured images. The flow velocity of the ions is a free parameter in the solution of the continuity equation; the value that minimizes the difference between the modelled and observed images is the extracted flow velocity at each time step. We demonstrate the method with an example event during the passage of a brightening arc feature, lasting about 10 s, in which the inferred electric fields vary between 20 and 120 mV m$^{-1}$. These inferred electric fields are compared with SuperDARN measurements, which have an average value of 30 mV m$^{-1}$. An excellent agreement is found in magnitude and direction of the background electric field; an increase in magnitude during the brightening of the arc feature supports theories of small scale auroral arc formation and electrodynamics.

## 1 Introduction

Horizontal electric fields close to dynamic aurora are a core building block of the electrodynamic system that links the ionosphere with the magnetosphere. The relation between these ionospheric electric fields and aurora is the subject of many studies





at both large and small scales and using many instrumental techniques. Rocket and satellite observations provide in-situ measurements of electric fields at sub-millisecond temporal resolution, but the rapid motion of the observing instrument means that such observations are typically interpreted as measurements of the spatial, not temporal, variation of the electric field. Marklund et al. (1982) studied the electrodynamics of an auroral feature, 20 km in width and aligned in the east-west direction,

using rocket-borne electric field measurements. These observations revealed a strong, predominantly northward electric field directed toward the southern boundary of the auroral feature. At smaller scales, Marklund et al. (1994) reported regions of very narrow ($\approx$1 km) and intense ($\approx$1 V m$^{-1}$) diverging electric fields observed by the Freja satellite, linking these fields to east-west aligned regions of dark lanes within bright aurora. Coincident ground based observations are extremely rare, since the satellite must travel through the local magnetic zenith for a precise comparison to be possible. To overcome this limitation,

? have combined *in situ* rocket and satellite data with Poker Flat Incoherent Scatter Radar (PFISR) and ground-based imagery to construct the temporal 2D pattern of ionospheric flow at the boundaries of quiet and stable arc structures. The images are used to identify the position of the arc boundary, while the *in situ* and radar data are extrapolated to replicate the 1D trajectory, and/or the radar slices through the arc. The resulting flow pattern was found to be consistent with a background flow outside the region of the arc, with a perturbation (or rotation) superimposed in the region of the arc.

In ground based radar observations electric fields are not measured directly, but are inferred from the observed plasma velocities using the $\mathbf{E} \times \mathbf{B}$ drift. They have typical temporal resolution of tens of seconds. Aikio et al. (2002) used the mainland EISCAT radars with complementary optical observations to investigate the electric fields of stable auroral arc systems, between 15 and 50 km in size, as they drifted through the radar beam, finding electric fields of up to 100 mV/m in strength directed perpendicular to the optical auroral arc. Lanchester et al. (1996) studied the electrodynamics of a system of narrow auroral

arcs, each with a width of about 1 km, using the EISCAT radars and optical instruments, with the highest possible temporal resolution for the radar of 3 s and with the camera resolution of 0.04 s. Large electric fields of up to 400 mV m$^{-1}$ were inferred, mainly pointing towards the arc. Further, the electric field strength increased when the brightness of the arc increased; however, the optical brightness increased more than the electric field strength. Due to differences in temporal resolution of the observations, these results indicate that even larger electric fields are likely to exist at timescales shorter than the resolution of

the radar. The magnitude of the auroral electric fields is a key ingredient for measuring parameters such as the heat input from auroral precipitation and currents.

Another method that has potential for deducing electric fields from the flow of optical emissions under auroral conditions is described in Blixt et al. (2006). This optical flow analysis was applied to two auroral events using narrow field video sequences in white light containing small-scale structure and dynamics. The particular algorithm that was applied provides estimates

of the flow field of the light, as well as a quantitative measure of where the constraints in the flow model break down. The robust optical flow estimator performed well for regions of turbulent motion, but less well for shear flow. The work illustrates the challenges of resolving the flow perpendicular to brightness gradients, and the difficulties of dealing with violations of smoothness and data conservation constraints, all of which are important in the present work.

The method we present here estimates plasma drift using optical measurements at high temporal and spatial resolution in a

region close to dynamic auroral features. The uniqueness of the method is the way that it uses three different wavelengths in





the aurora to solve the continuity equation for the metastable oxygen ion. This $O^+(^2P)$ ion emits as it drifts under the influence of electric fields close to an auroral arc. The other unique feature of the results is that the optical measurements are close to the temporal resolution of the cameras at 0.1 s, and at a spatial resolution of 100 m, which is close to the limit of auroral structure widths (Sandahl et al., 2011). The instrument used is the Auroral Structure and Kinetics (ASK) instrument, which
was designed for the purpose of measuring plasma flows in a small 3.1°×3.1° field of view around the magnetic zenith.

Dahlgren et al. (2009) were the first to use observations of auroral brightness from the ASK instrument to estimate plasma flow velocities, from which the ionospheric electric field was inferred. They tracked the motion of the afterglow from the metastable $O^+$ ions produced by auroral precipitation in narrow arcs by assuming a fixed emission height for the afterglow. They inferred electric fields of a few tens of $mV\,m^{-1}$ as an auroral event subsided; however, such tracking was not possible
during the main brightening as the motions of the source and the plasma could not be separated.

The present method, referred to as the "flow model", overcomes the limitations of the above study through the following steps.

(1) It accurately separates the prompt emissions which occur at the point of impact of precipitation from the emissions of the metastable ions which may have moved from the location of the source.

(2) It solves the continuity equation for the $O^+$ ions at all heights in the 3D region surrounding the zenith direction. This solution requires an accurate estimate of the magnetic zenith position in the images, and must account for perspective effects in the region away from the zenith.

(3) By using the flow velocity as a free parameter, the solution determines how the three-dimensional distribution of the $O^+$ ions evolves during periods of auroral electron precipitation.

(4) From this time evolving distribution, modelled images of emission are projected onto the image plane on the ground (Rydesäter and Gustavsson, 2001; Tuttle et al., 2014).

(5) The flow velocity is extracted by finding the velocity that minimizes the difference between the modelled and observed images.

## 2   Instrumentation and Observations

### 2.1   The ASK instrument: three emissions

The Auroral Structure and Kinetics instrument (ASK) is a ground-based optical instrument consisting of three low-light imagers capable of resolving the structure and dynamics of the aurora at resolutions of 20 m and 0.025 s. In the present study, each imager had a field of view of 3.1×3.1° (equivalent to about 5×5 km at 100 km altitude). Each camera has a specially selected narrow passband filter to isolate chosen emissions from the total auroral brightness. All cameras are synchronised and aligned
centered on the magnetic zenith which is the only direction in which the true width of an auroral feature can be measured accurately. Perspective effects are critical in this small region; a new method is applied which allows emissions that are off the zenith field line to be included in the flow model.





The first imager (ASK1) isolates emissions from several bands of the $N_2$ 1PG electronic band system (Ashrafi et al., 2009) using a passband 14.0 nm wide centred at 673.0 nm. These emissions are due to excitation of $N_2$ molecules by precipitating electrons, and exhibit little dependence on the energy of the precipitating electrons. Therefore the brightness of this emission can be used to estimate the energy flux of the electron precipitation (Lanchester et al., 2009). There are no other known auroral emissions in this wavelength region.

The second imager (ASK2) isolates emissions from the metastable $O^+(^2P)$ ion using a passband 1.0 nm wide centred at 732.0 nm. These emissions are from transitions between the $^2P$ and $^2D$ states, which are discussed further in section 3.3. Auroral emissions observed by this imager are indicative of low energy ($< 1$ keV) precipitation (Dahlgren et al., 2008, 2009). However, there is some contamination from the (5,3) band of the $N_2$ 1PG band system and hydroxyl airglow. The $N_2$ contamination is removed using the method of Spry et al. (2014), and the hydroxyl contamination is removed by background subtraction.

The third imager (ASK3) isolates emissions due to the transition between the $^5P$ and $^5S$ states of neutral oxygen using a passband 1.5 nm wide centred at 777.4 nm (Lanchester et al., 2009). Two processes produce the upper excited state: electron impact excitation of atomic oxygen and dissociation of molecular oxygen. Altitude variations in the abundances of atomic and molecular oxygen cause the excitative process to be more sensitive to low energy precipitation and the dissociative process to be more sensitive to high energy ($> 1$ keV) precipitation. This energy dependence results in emission from all precipitation energies, but it is more responsive to low energy precipitation than the 673.0 nm emission. There are no contaminating emissions beyond the background brightness.

The observations presented here were obtained when ASK was co-located with the EISCAT Tromsø radar facility at Ramfjordmoen, Norway (69.6° N, 19.2° E) and ASK was observing in the direction of the local magnetic zenith. All data are dark and flat-field corrected, background subtracted and intensity calibrated using star fluxes. Measured intensities of tens of stars per image are compared with spectral irradiances in absolute units from tabulated values (Gubanov et al., 1992; Cox, 2000; Grubbs II et al.).

## 2.2 Energy and Energy Flux

The different sensitivities of emission at 673.0 nm ($N_2$) and 777.4 nm (O) to the energy of precipitation means that the ratio of O/$N_2$ brightnesses provides an estimate of the characteristic energy of the electron precipitation. The Southampton auroral model is described in more detail in Lanchester et al. (2009), in which the method of using this ratio was tested during an auroral event measured with the ASK instrument and with incoherent scatter radar. The characteristic energy and energy flux are parameters needed for the production term in the solution of the continuity equation of the $O^+$ ions.

The 1D auroral model is time-dependent and solves the electron transport equation (Lummerzheim and Lilensten, 1994) at each time step, resulting in output height profiles of auroral ionization, excitation, and electron heating rates. These are the inputs to the ion chemistry and energetics part of the model, in which the time-dependent coupled continuity equations for all important positive ions and minor neutral species are solved along with the electron and ion energy equations (Lanchester et al., 2001). Initial conditions relevant to each event include estimates of neutral densities from the Mass Spectrometer Incoherent Scatter (MSIS) model (Hedin, 1991) and solar and geomagnetic indices, such as the $F_{10.7}$ solar radio flux and the $A_p$ index.





The cross-sections used are those described in Ashrafi et al. (2009) for the ASK1 ($N_2$) emission and Julienne and Davis (1976) for the ASK3 (O) emission. A filter transmission factor is applied to each emission. In the case of the $N_2$ 1PG (4,1) and (5,2) bands, the transmission factor through the ASK1 filter has a value of 0.72 determined using synthetic spectra and the filter transmission. For the OI multiplet the transmission factor is 0.70. Modelled emission brightnesses are obtained by height integrating the emission rate profiles.

Such modelled emission brightnesses are combined with measured brightnesses of 673.0 nm and 777.4 nm, from ASK1 and ASK3 respectively, to estimate the energy and flux of the electron precipitation in the magnetic zenith (Lanchester et al., 2009; Lanchester and Gustavsson, 2012). The ratio of the modelled emission brightnesses is determined as a function of peak energy, under conditions appropriate for the time and date of the event. The peak energy of the electron precipitation is then estimated from the ratio of the observed emission brightnesses. The energy flux is estimated from the ASK1 $N_2$ 1P emission brightness. A conversion factor of 250 Rayleighs per $\mathrm{mW\,m^{-2}}$ is used.

### 2.3 SuperDARN

The Super Dual Auroral Radar Network (SuperDARN) of pairs of HF radars operates in overlapping regions mainly at high latitudes (Chisham et al., 2007). In the present work we use primarily the two CUTLASS radars that overlap the field of view of the optical and EISCAT instruments, i.e. the radars at Pykkvibaer, Iceland, and Hankasalmi, Finland. Plasma density irregularities in the F region ionosphere backscatter the HF waves emitted by the radar and the Doppler shift received gives the line-of-sight velocity component of the E×B drift. During the period of interest, the radars were operating in Common Time mode, in which each radar performs a sweep of its field of view every minute. Each sweep is formed by sequentially scanning 16 beams, each of which is separated in azimuth by 3.24 degrees. Each beam is separated into 75 range gates, each with a length of 45 km. Where measurements from two radars overlap, the data can be "merged" to give the 2D horizontal flow velocity at these heights (Ruohoniemi and Baker, 1998).

### 2.4 Observations

The presented event is from 9 November 2006, during a time of increased auroral activity caused by the Earth entering a fast solar wind stream with negative $B_z$ at about 20 UT. At 21:25 UT bright, structured and dynamic aurora was observed in the magnetic zenith. Figure 1 is an overview of the event. Panels (a)–(c) show false color images of the observed auroral forms in the three wavelengths, at selected times during a 15 s interval. In (a) and (c) the 673.0 nm ($N_2$) and 777.4 nm (O) images, which measure the presence of high energy precipitation, start with a diffuse aurora with no distinguishable features. During a 10 second interval a north-south aligned filament becomes structured and moves through the magnetic zenith. Panels (d)–(f) are stack plots of west-east slices across the feature through the magnetic zenith for all three wavelengths. The position of the slice is marked on the first image of panels (a)–(c). At 21:25:04 UT the aurora brightens over a two second interval, increasing from 5 kR to 12 kR at 673.0 nm and from 2 kR to 6 kR at 777.4 nm, peaking at 21:25:06.5 UT. The auroral brightness then decreases to initial levels at 21:25:08 UT before further abating over the next few seconds until the feature is no longer distinguishable.





The 732.0 nm (O$^+$) brightness in Fig. 1(b) does not exhibit the same behaviour as that at 673.0 nm and 777.4 nm. After the

intensification at 21:25:06.5 UT, the 732.0 nm brightness does not fall as rapidly as that of either the 673.0 nm or 777.4 nm emissions, as a result of the metastable nature of the O$^+$($^2$P) ion. As seen in the 732.0 nm slices in Fig. 1(e), these emissions continue for up to 5 seconds after the prompt emissions diminish, with an eastward component of the motion of the emission (west is at the top). This motion is not caused by motion of the source, which is shown in the prompt emissions at 673.0 nm and 777.4 nm, but is a result of the $\mathbf{E}\times\mathbf{B}$ drift of the metastable O$^+$($^2$P) ions caused by horizontal electric fields in the ionosphere.

## 3   Flow velocity modeling

### 3.1   Position of the magnetic zenith

To determine accurate estimates of the energy and flux of the precipitation, the exact position of the zenith within the images must be known. Models of the geomagnetic field, such as the International Geomagnetic Reference Field (IGRF), are often used to calculate the position of the magnetic zenith at auroral altitudes. However, such models do not account for dynamics

of the magnetic field under auroral precipitation conditions. Variations in the direction of the magnetic field of greater than one degree have been observed (Maggs and Davis, 1968; Whiter, 2008), which is significant for narrow field of view imagers such as ASK. The position of the magnetic zenith can be estimated if rayed structures are present in auroral observations. These rays are spatially confined perpendicular to the field, but can extend several hundreds of kilometers parallel to the field. Maggs and Davis (1968) first used such a method in their seminal paper on the width of auroral structures, to estimate the

location of the radiant point, or magnetic zenith, and found it was within an ellipse of 1 degree by 2 degrees.

Rayed structure is present at times during the interval studied here. Figures 2(a), 2(b) and 2(c) show auroral forms that exhibit rays 1.6 s before and 3.8 s and 8.5 s after 21:25 UT, respectively. A line is drawn manually along each ray; the start and end points of this line are indicated by the asterisks in the figure. The line is extended across the image and the minimum distance from each pixel to the line is calculated. The location of the magnetic zenith within the images is at the pixel which minimises

the sum of the squares of the distances between that pixel and each of the ray lines. The error at each pixel is shown in Fig. 2(d) and the estimated position of the magnetic zenith is found to be at the minimum error. In Figs. 2(a), 2(b) and 2(c) the double circles indicate the magnetic zenith obtained using the ray method presented here; the single circles indicate magnetic zenith calculated using the IGRF model. The difference is of the order of one degree, which makes a significant difference to the interpretation of the images. The azimuth and elevation angles of the magnetic zenith are given by the azimuth and elevation

of the line of sight of the pixel at the position of the magnetic zenith. This pixel is defined as the zenith pixel, $(u_z, v_z)$.

### 3.2   Correction for perspective

At all angles away from the magnetic zenith, the emissions from each camera at a given pixel are no longer on the same field line; this perspective effect must be accounted for when using images, even within a few degrees of the zenith position. Along an individual field line, at a given time, emissions at all heights result from a single electron energy spectrum that precipitates





down through the atmosphere. Under this constraint, we use simple geometrical arguments to correct for perspective effects when estimating the energy of electron precipitation in the region close to the magnetic zenith. Figure 3 depicts a situation when aurora occurs along a magnetic field line that is some perpendicular distance, $d$, away from the magnetic zenith. Emission rate profiles produced by electrons that precipitate along that field line are shown on the right of Fig. 3, in idealised form, for the emissions observed by ASK1 ($N_2$) and ASK3 (O). The peak heights for each profile are marked by dashed lines. The pixel

lines of sight through the positions of these peak heights of emission subtend angles $\theta_1$ and $\theta_3$ with the magnetic zenith for the emissions observed by ASK1 and ASK3, respectively.

Figure 4 is a representation of an image in ASK1. We define the image distance between the zenith pixel, $(u_z, v_z)$, and the pixel whose line of sight passes through the height of peak emission to be $n_1$ in the ASK1 image, and similarly to be $n_3$ in the simultaneous ASK3 image. The unit vector $\hat{\underline{r}}$ from any given pixel $(u, v)$ toward the zenith pixel is given by:

$$\hat{\underline{r}} = \frac{(u_z - u)\hat{\underline{u}} + (v_z - v)\hat{\underline{v}}}{\sqrt{(u_z - u)^2 + (v_z - v)^2}} \tag{1}$$

where $\hat{\underline{u}}$ and $\hat{\underline{v}}$ are unit vectors in image co-ordinate directions, and the values of $(u_z - u)$ and $(v_z - v)$ are small displacements. As emissions observed by ASK3 originate from a slightly higher altitude than those observed by ASK1, features in ASK3 will appear closer to the zenith pixel than features in ASK1. We therefore use the image coordinates of the ASK1 image as a reference and the perspective correction is applied to the ASK3 image. The denominator in Eq. (1) is then the image distance

$n_1$. We define

$$n_s = n_1 - n_3 \tag{2}$$

which is the image distance that the position of the ASK3 peak emission appears shifted toward the zenith relative to the position of the ASK1 peak emission. This shift is the perspective effect that is corrected by the following geometrical argument.

The variation of image distance with angle is linear and obeys the following relation:

$$\frac{\theta_T}{n_T} = \frac{\theta_1}{n_1} = \frac{\theta_3}{n_3} \tag{3}$$

where $n_T$ is the total image distance and $\theta_T$ is the total field of view of the observed image. The angles in Eq. (3) are eliminated, using trigonometry and the small angle approximation, to yield the following relation between the image shifts and the altitudes of peak emission, $h_1$ and $h_3$:

$$n_3 = n_1 \frac{h_1}{h_3} \tag{4}$$

The altitudes of peak emission are obtained from an initial estimate of the energy that is obtained using the methods described in Sect. 2.2. Equation (4) can be combined with Eq. (2) to yield

$$n_s = n_1 (1 - \frac{h_1}{h_3}) \tag{5}$$

Combining Eq. (5) with the expression for $\hat{\underline{r}}$, the displacement of the shift toward the zenith, $\underline{r}_s$ is found:

$$\underline{r}_s = n_s \hat{\underline{r}} = (1 - \frac{h_1}{h_3})\{(u_z - u)\hat{\underline{u}}) + (v_z - v)\hat{\underline{v}})\} \tag{6}$$





where shifts along the image co-ordinate directions are given by:

$$\Delta u = (1 - \frac{h_1}{h_3})(u_z - u) \tag{7}$$

and

$$\Delta v = (1 - \frac{h_1}{h_3})(v_z - v) \tag{8}$$

Therefore, to account for perspective effects, the following ratio of brightnesses should be used for each pixel $(u,v)$:

$$R(u,v) = \frac{B_1(u,v)}{B_3(u + \Delta u, v + \Delta v)} \tag{9}$$

where $B_1(u,v)$ is the brightness in the ASK1 image at pixel $(u,v)$ and $B_3(u + \Delta u, v + \Delta v)$ is the brightness in the ASK3 image at pixel $(u + \Delta u, v + \Delta v)$. By taking this ratio at every pixel in the ASK1 images, a map of the peak energy of the electron precipitation is produced along all field lines in the image.

One such map of peak energy across the ASK field of view at the time of the brightening at 21:25:06 UT is shown in Fig. 5.

The black line is the 4 kR brightness contour from the observed 673.0 nm emission at this time. The estimates of the energy of the precipitating electrons are approximately 1 keV along the edge of the feature closer to zenith, increasing to 5 keV on the edge further from zenith, and corresponding to the lower border of the arc. The changes in the estimated energies when the perspective correction is applied are an increase of approximately 0.2 keV (20%) in the region closer to the zenith, and a reduction of 0.5 keV (10%) further from the zenith at lower heights, consistent with the geometry of the arc.

### 3.3  The O$^+$($^2$P) ion continuity equation

The dynamics of O$^+$($^2$P) ions are governed by a continuity equation, which has terms for production, quenching, emission, drift and diffusion:

$$\frac{dn}{dt} = q - \sum_i \alpha_i n_i n - \sum_j A_j n - \nabla n \cdot \mathbf{v} - n \nabla \cdot \mathbf{v} - D \nabla^2 n \tag{10}$$

where $n$ is the density of O$^+$($^2$P) ions, $q$ is the production rate of O$^+$($^2$P) ions, $n_i$ is the density of quenching species $i$, $\alpha_i$ is the rate coefficient for quenching by species $i$, $A_j$ is the Einstein coefficient for radiative transfer from the $^2$P state to state $j$,

$\mathbf{v}$ is the velocity of O$^+$($^2$P) ions and $D$ is the diffusion coefficient. The contribution each term makes in Eq. (10) is described below, as well as how each term is obtained in order to solve for the three dimensional distribution of O$^+$($^2$P) ions.

The first term on the right is the production of O$^+$($^2$P) ions, which occurs by impact ionization of neutral atomic oxygen by precipitating electrons through the following process:

$$e^- + O \longrightarrow 2e^- + O^+(^2P, {}^2D, {}^4S) \tag{11}$$

with 18% of O$^+$ ion production into the $^2$P state (Rees, 1982). The $^2$P state is further split into J$_{1/2}$ and J$_{3/2}$ angular momentum

states. Production rates of O$^+$($^2$P) ions are obtained using a combination of optical observations and modelling as described





in Sect. 2.2. First, estimates of the peak energy are found using brightness ratios of emissions from $N_2$ 1P (673.0 nm) and O (777.4 nm), taking into account the perspective effects across the image as described in Sect. 3.2. The resulting energies (e.g. shown in Fig. 5) and fluxes (from 673.0 nm brightness) are used as input to the Southampton auroral model for each time step during the event in order to give the production of $O^+$ ions along each field line in the field of view. More details of the chemistry of the $O^+(^2P)$ ions and modeling of $O^+$ densities using the Southampton auroral model can be found in Dahlgren et al. (2009).

The second and third terms on the right of Eq. (10) are the two loss processes affecting the $O^+(^2P)$ ion: quenching and emission. Quenching is the dominant loss process at higher atmospheric densities, and hence lower altitudes. We use the rate coefficients for quenching by electrons given by Rees (1989), and rate coefficients for quenching by oxygen and nitrogen obtained by Stephan (2003). Emission occurs when $O^+(^2P)$ ions de-excite by spontaneously emitting a photon; there are no stimulated emissions. There are two radiative paths, $(^2D)–(^2P)$ and $(^4S)–(^2P)$, through which there are six possible transitions. These transitions form emission doublets at 733 nm, 732 nm and 247 nm, with only the 732 nm doublet observed by the ASK2 filter. Einstein coefficients for these transitions are found in a study made of the $O^+$ doublets by Whiter et al. (2014). The 732 nm doublet emission has contributions from both the $J_{1/2}$ and $J_{3/2}$ states, which means that Eq. (10) must be solved for each angular momentum state. In isolation, the emission and quenching terms can be inverted to obtain altitude dependent effective lifetimes for the two angular momentum states of the $O^+(^2P)$ ion (Dahlgren et al., 2009). These lifetimes are calculated using:

$$\tau(z) = \frac{1}{\sum_i \alpha_i n_i(z) + \sum_j A_j} \tag{12}$$

with the altitude dependence arising from the density profiles of the quenching species.

The fourth and fifth terms on the right hand side of equation (10) arise due to the flux term, $\nabla \cdot (n\mathbf{v})$, of the continuity equation. Rather than solving these terms explicitly to determine $\mathbf{v}$, the velocities in the modelled region are parameterized. This parametrization is discussed further in Sect. 3.4.

Finally, the diffusion term can be neglected for the $O^+(^2P)$ ion, because collisions that would ordinarily redistribute the thermal motion of the ion instead cause the ion to quench. Perpendicular to the magnetic field, strong density gradients may exist. At high altitudes, where quenching is negligible, these gradients are maintained by the magnetic field.

### 3.4 The flow model

Equation (10) is solved in a volume 30 km by 30 km by 410 km with the long axis parallel to the magnetic field. The volume is positioned such that the field of view of ASK is fully enclosed by the volume at all altitudes. The spatial resolution is 200 m along all dimensions. The only free parameters in Eq. (10) are those that parameterize the velocity of the ions. There are several possible methods by which this velocity can be parameterized. Here, one of the simplest methods is used: a uniform velocity across the whole 3-D volume. This velocity is broken down into components, with one parallel and two perpendicular to the magnetic field. The number of free parameters is further reduced by neglecting ion motions parallel to the magnetic field. Therefore we search for two free parameters, the components of the ion velocity perpendicular to the magnetic field.



The optimal free parameters, $P$, are searched for by minimizing an error function. The error function used here is the sum of the square of the difference between the observed and modelled images, and is given by:

$$err(P) = \sum_{u,v} [I_{obs}(u,v) - I_{mod}(u,v,n(\boldsymbol{r},P))]^2 \tag{13}$$

where $I_{obs}(u,v)$ is the observed brightness at pixel $(u,v)$ and $I_{mod}(u,v,n(\boldsymbol{r},P))$ is the modelled brightness at pixel $(u,v)$.

The modelled brightnesses are obtained using a forward model $f$:

$$I_{mod}(u,v,n(\boldsymbol{r},P)) = f(n(\boldsymbol{r},P)) \tag{14}$$

where $n(\boldsymbol{r},P)$ is the density of $O^+(^2P)$ ions, calculated from Eq. (10) using the trial free parameters, $P$, at the position $\boldsymbol{r}$. The forward model uses the blob-based dot projection method of Rydesäter and Gustavsson (2001) to project emission from the 3-D distribution of $O^+(^2P)$ ions to an image plane on the ground, forming modelled images of the 732.0 nm brightness. To allow comparison with the observed brightness, the modelled image is converted to Rayleighs using a calibration image with a

uniform brightness of 1 Rayleigh. The calibration image is formed by applying the forward model to a volume with a uniform column emission rate of $10^{10}$ photons $\mathrm{m}^{-2}\mathrm{s}^{-1}$, which is the definition of a Rayleigh (Hunten et al., 1956).

## 4   Results

The flow model is run for a 15 second interval, starting at 21:24:57.50 UT, that includes times before, during and after the arc brightening. A timestep of 0.1 s is chosen, i.e. half the resolution of the ASK measurement, which is sufficient to resolve the

dynamics in this event. For the first five seconds the model is run without the optimization, and using a plasma velocity of zero, to generate an initial distribution of $O^+(^2P)$ ions to be tracked. In the remaining ten seconds the optimization is active and the model searches for the velocity at each timestep using the methods described above. Figure 6 is a sequence of modelled images at 0.5 s cadence of the 732.0 nm emission from the convecting distribution of $O^+(^2P)$ ions. The corresponding observations of the 732.0 nm emission (background subtracted) are shown in Fig. 7, with the modelled brightness contours superimposed. The

modelled images match the structure of the measured images well, while the brightness of the model images is about half of those observed.

Figure 8(a) shows the recovered velocity vectors for this same interval at 0.1 s cadence. Time in seconds is shown as color, with the time when the arc brightness increases indicated by the shaded region on the color bar. The velocities all have southward and eastward components, with a rotation from east to south, albeit with some variation. Both before and after the

feature brightens, the velocities are between 0.4 and 1.2 $\mathrm{km\,s}^{-1}$. When the feature intensifies the velocities increase, to a peak velocity of 2.4 $\mathrm{km\,s}^{-1}$. Figure 8(b) shows an equivalent pattern in the electric fields, which are inferred from the fact that the velocity indicates E×B drift at the altitude of peak emission. The magnitude of the magnetic field at 200 km altitude within the ASK field of view is calculated from the IGRF-12 geomagnetic model. The inferred electric fields are found to have southward and westward components and have magnitudes of between 20 and 120 $\mathrm{mV\,m}^{-1}$.

At the time of these observations, the SuperDARN radars at Hankasalmi and Pykkvibaer were also measuring ionospheric flows above Svalbard, so a direct comparison between the modelled and measured velocities is possible. Line of sight plasma





velocities from the two radars have been merged at two minute resolution to give a measure of the larger-scale flow. Figure 9 shows the magnitudes (length and color) and directions of the plasma velocities between 21:24 UT and 21:26 UT. The small black square is the approximate size and position of the ASK field of view at 200 km height. Table 6 gives the four merged

velocities at positions closest to that of ASK, given by their magnetic latitude and longitude, and labeled 1–4 in the figure. The position of the field line above ASK at the height of the emissions is also included. These merged velocities all have southward and eastward components and have a narrow spread in direction. The average plasma velocity (600 m s$^{-1}$) and electric field (30 mV m$^{-1}$) vectors from the four SuperDARN measurements are shown in Fig. 8 by the thick black dashed lines, showing clear agreement in direction, but with smaller magnitudes.

## 5   Discussion

To resolve the full electrodynamics of auroral arc formation, high temporal and spatial resolution is a crucial requirement. It impacts theories of auroral acceleration in the inner magnetosphere since resulting field-aligned currents must close in the dynamic auroral ionosphere, where electric fields are measured. The high time resolution of the present method of 0.1 s places constraints on theories to resolve the dynamic nature of electric fields close to auroral precipitation, and the intimate

connection between the two. Spatial resolution is also of importance, and although our measurements give optical resolution of 10s of metres, the present model is tested by using the simplest form of parametrization of the plasma velocity, which is a uniform flow perpendicular to the magnetic field. Therefore the present determination of the plasma flow is an estimate over a volume corresponding to the optical field of view, which is $5 \times 5$ km at 100 km height, an order of magnitude smaller than the volume for the coherent radar measurement.

It is useful to set the small scale nature of this result within the large scale auroral environment. Figure 10 shows the fitted SuperDARN vectors across auroral latitudes. At this time the ASK instrument is close to magnetic midnight, where there is a clear signature of ion flows towards the south east, as the dawn cell has expanded under the influence of a positive $B_y$ component of the IMF. The magnitudes of the SuperDARN velocities measured in the region over ASK are toward the lower end of the range of velocities obtained from the optical model during the passage of the arc, and correspond to the times when

the brightness of the auroral feature was not enhanced. Due to their lower cadence, the SuperDARN vectors are more likely to be representative of the background plasma flow, rather than the arc-related enhancement in flow obtained from the high cadence optical measurements.

The sub-second electric fields, calculated during the 10 s passage of the arc through the field of view of the cameras, are enhanced when the arc is brighter. This result is important for theories of auroral electrodynamics, in particular how

the ionospheric electric fields link to processes further out where acceleration of electrons is taking place. The theory of Birk and Otto (1996) as presented in Lanchester et al. (1997) uses a three-dimensional multi-fluid MHD model to simulate the spatial variation of the ionospheric plasma velocity close to a narrow ($< 1$ km) and dynamic arc filament, similar to that observed in the present event. The simulation includes a magnetic and velocity perturbation at its upper boundary in the inner magnetosphere, which generates field-aligned current sheets. A field-aligned electric field is generated by a resistive term in





Ohm's law if the current density exceeds a threshold value. A small-amplitude perturbation is applied to initialize the formation

of the acceleration region. The resulting plasma velocities in the ionosphere, which map to a similar sized region as our field-

of-view of a few km$^2$ at auroral heights, are found to be mainly tangential to the arc filaments, with inferred electric fields

pointing towards regions of enhanced potential, and with increased magnitudes where the arc is brightest. The net electric field

at any time will be the sum of the background electric field and the electric field due to the feature. This combination should

result in changes to both the magnitude and direction of the electric fields across the field of view. In our result, we assume

that velocity, and hence electric field, is uniform throughout the modelled volume. Such an assumption is therefore unable to

account for the very small spatial variations of electric fields, such as those that may be generated on either side of the $< 1$ km

scale auroral feature. A future step is to apply more complex parametrizations, e.g. a shear flow across the arc, similar to the

above simulations.

The comparison of the derived velocities with the SuperDARN velocities reinforces the above interpretation. Care must be

taken, however, as the SuperDARN velocities are obtained by merging individual line of sight vectors from two different radars,

which are taken from different instants within the two minutes (and which fall outside the period covered in Fig. 8, and so are

not simultaneous with the ASK measurements). We also note that the SuperDARN velocities may be underestimated as a result

of uncertainty in the refractive index as shown by Gillies et al. (2012). This effect is likely to be of the order of $10\%$. However,

the SuperDARN observations are an excellent measure of the background plasma velocities on timescales much longer than

the modelled interval (10 s), as shown by the close agreement between the SuperDARN velocities and the modelled velocities

before and after the feature intensifies.

The presented method for determining dynamic electric fields includes several steps, all of which have inherent uncertainties

which must be evaluated, in order to have confidence in the high cadence vectors of Fig. 8. An important parameter in the

modeling of optical images is the magnetic field direction at auroral altitudes within the volume enclosed by the ASK field of

view. Field-aligned currents within auroral features cause perturbations to the background magnetic field, which means that the

position of the magnetic zenith may vary. In the present event, lines which pass through field-aligned rays are drawn manually,

so there are uncertainties in position of not more than five pixels in selected points on each ray line. However, all ray lines pass

within two pixels of the recovered position of the magnetic zenith, suggesting this uncertainty is small in the event presented

here. The rays used to reconstruct the position of the magnetic zenith are separated by up to 10 seconds. The reconstruction

assumes that during this interval, the position of the magnetic zenith does not vary. It is clear from Fig. 2 that the zenith position

is indeed a much better estimate than is given by the IGRF. The lower panels of Fig. 11 also provide evidence for the improved

zenith estimate; there is a ray visible in the prompt O emissions, which is aligned well with the zenith. The direction of the

magnetic zenith is critical for the application of the geometric correction for perspective in this small region within the images.

The main assumption used in the correction for perspective is the height of the peak emissions found from modeling, using

an initial estimate of the energy from the uncorrected brightness ratio. For the region around the lower border of the arc the

energy is 5 keV; the difference in the height of peak emissions from $N_2$ and O is a few hundred metres, resulting in small

or negligible positional shifts, no matter how far the feature is from the zenith. In regions where the energy is 1 keV, the

difference in peak emission height is a few kilometres, which could cause significant perspective effects. For the present event,





the regions of the bright feature where the energy is low are close to the zenith, which reduces the magnitude of the required perspective-correction. The correction as applied is consistent with the observed geometry, and hence improves the accuracy of the flow model in the 3D volume around the zenith.

The main uncertainty in the optical measurements which could affect the ratio estimation, and hence the absolute values of the peak energy (after the application of the perspective correction), is the intensity calibration of the imager data. The ASK

data are calibrated by comparing measured star brightnesses with tabulated values. Any effect from scattering into each pixel is removed in the background subtraction, which is taken from an interval of no aurora in the field of view 20 minutes before and after the event. Therefore any changes in the background brightness between the period devoid of emissions and the event, such as haze or thin cloud in the field of view, are not included. For bright features, uncertainties in the brightness of the background have a negligible effect on the estimates of energy and energy flux. However, for features with a lower emission brightness,

such as diffuse aurora, energies and fluxes estimated would be more uncertain. For the bright O and $N_2$ emissions measured in this event, the uncertainty in the absolute intensity calibration has been quantified at 20%.

Uncertainties arising from the auroral model include the assumption of an input neutral atmosphere, here taken from MSIS. It is known that the oxygen density may be significantly reduced from the model profiles. However, as shown in Fig. 3 of Lanchester and Gustavsson (2012), the effect on the estimation of energies is more marked for low energy precipitation, partic-

ularly lower than 1 keV. The present event is dominated by higher energies, and the effect is estimated to be less than 10% for the 1 keV electrons, and with negligible effect for energies of 5 keV. The accuracy of the model emission rates can be checked by a direct comparison between the observed images from ASK1 (673.0 nm) and ASK3 (777.4 nm) and modelled images. The latter are formed from emission rates obtained in the 3D region of the flow model, which are projected to an image plane on the ground, at the ASK location, using the same blob-based dot projection method that was used in Section 3.4 for the $O^+$

emission. Figure 11 shows this comparison. The black lines delineate the brightness extent of the discrete auroral feature in the respective observations (4 kR for 673.0 nm and 2 kR for 777.4 nm). The position, structure and brightness of the discrete auroral feature in the observed and modelled images show excellent agreement. Such agreement could not be achieved if the energies and fluxes of the modelled precipitation were not close to the observed values.

The uncertainty in the peak energies will have a negligible effect on the $O^+$ production term, but the shape of the input

spectrum may affect this term. The brightness of the modelled $O^+$ is less than that measured by about 50%. The most likely reason is that the model does not include a low energy contribution in the present runs. Since such a low energy tail is an arbitrary addition, we have chosen to use the Gaussian shaped spectra unless there is evidence (e.g. radar profiles) to support a different shape. (Testing of such variable shapes is the subject of a separate study.) Increasing the low energy input to the model would result in an increased brightness as well as a slight increase in the height of the peak $O^+$ emission. Therefore the

present result for the magnitude of the electric fields is likely to be an underestimate. The direction would not be affected.



## 6   Conclusions

The present work demonstrates a new method for estimating plasma flows around auroral features, using measurements from a multi-monochromatic imager, and modeling. The dynamic nature of the auroral event is captured at a resolution of 0.1 s, for a 10 s interval during which the arc passed through the magnetic zenith, a typical time span for dynamic aurora. Most
measurements to date of electric fields average over much longer intervals than 10 s and therefore miss dynamic changes. The agreement found in both magnitude and direction of the flow velocities with those measured by coherent radar gives confidence that the background flow is well captured. In this instance it is of magnitude about 0.6 km s$^{-1}$ in the south east direction. The increase in magnitude of the flows to 2.4 km s$^{-1}$, with equivalent electric field magnitudes of 120 mV m$^{-1}$ during the brightening of the arc feature, agrees with present theories of small scale auroral arc formation.

For the event presented, the simplest form of parametrization, that of a uniform flow perpendicular to the magnetic field, was applied through the modelled volume. The full 3D model as presented here addresses some of the issues of optical flow analysis algorithms as done in e.g. Blixt et al. (2006) (and references therein). Future work will test the confidence limits of the retrieved electric field under different auroral conditions, as well as investigate more complex parametrizations.

*Author contributions.* ST performed the analysis of the data presented, and was responsible for all the stages of the flow model code.
BL supervised the work during ST's doctorate. BG contributed significantly to elements of the flow model. NI designed and supervised the construction of the ASK instrument, with the first aim of solving the continuity equation for O$^{+}$ ions. DW supervised the running of the experiment and developed the auroral model, RF and ML provided analysis of SuperDARN data. BL prepared the manuscript with contributions from all co-authors, mainly ST.

*Competing interests.* The authors declare that they have no conflict of interest.

*Acknowledgements.* This work was supported by the Natural Environment Research Council of the UK (grant number NE/H024433/1). The ASK instrument has been funded by PPARC, STFC and NERC of the UK and by the Swedish Research Council. RCF was supported by STFC Ernest Rutherford Fellowship ST/ K004298/2 and Consolidated Grant ST/R000719/1. We thank the members of the Space Environment Physics group who set up and ran the instrument during the winter of 2006-7 (H. Dahlgren, O.-P. Jokiaho and J. M. Sullivan), who have contributed to fruitful discussions. SuperDARN is funded by national scientific funding agencies of Australia, Canada, China,
France, Japan, South Africa, United Kingdom and United States of America. The raw ASK data may be viewed in summary form at http://space.soton.ac.uk/data. They are stored on magnetic tapes and are made available upon request.



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



**Figure 1.** (a), (b) and (c) Images of the auroral brightness observed by ASK1 ($N_2$, $I_{673.0}$), ASK2 ($O^+$, $I_{732.0}$) and ASK3 (O, $I_{777.4}$) respectively. The asterisk is the position of the magnetic zenith and the dashed lines are the position and orientation of the slices below. (d), (e) and (f) Stack plots of west-east slices, 5 pixels wide, through images from ASK1, ASK2 and ASK3 respectively. The black lines link the images to the selected times. Logarithmic intensity scales are used to highlight both bright and faint features throughout the event.







**Figure 2.** (a), (b), (c) Images of the auroral brightness observed by ASK1 (673.0 nm). The black lines pass through rayed structures; the asterisks indicate the region of the line drawn manually. The single circle in each image is the position of the IGRF estimate of the magnetic field. The double circles mark the estimate of the zenith position from the four rays. (d) The sum of the squares of the distances between a given pixel and each ray line.



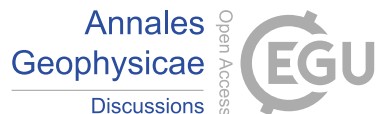

**Figure 3.** Geometry used to estimate perspective effects. Idealised emission rate profiles are shown on the right for ASK1 ($N_2$) and ASK3 (O).





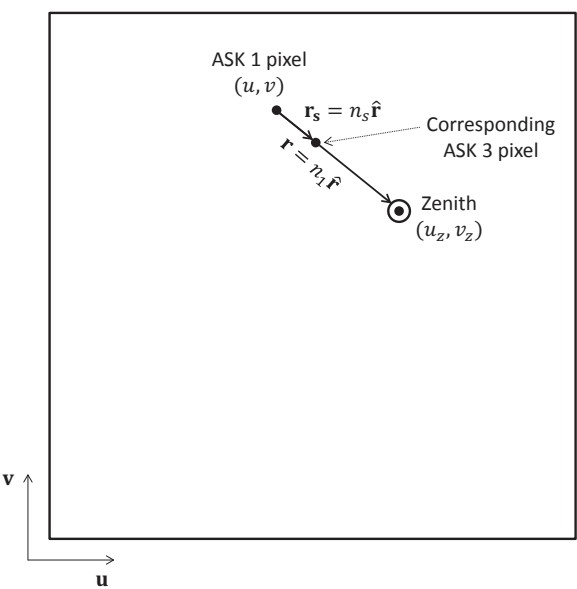

**Figure 4.** Representation of an ASK frame showing definition of image distance and shift.

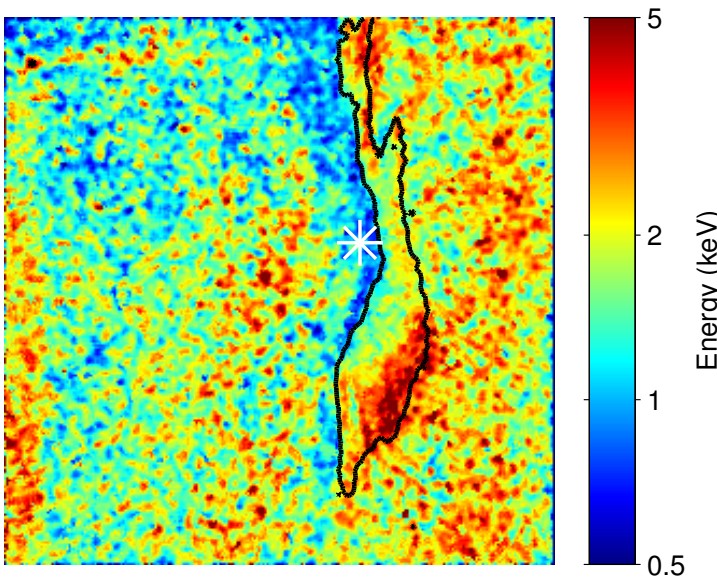

**Figure 5.** Perspective corrected estimate of energy across the ASK field of view. The white asterisk indicates the position of the magnetic zenith.



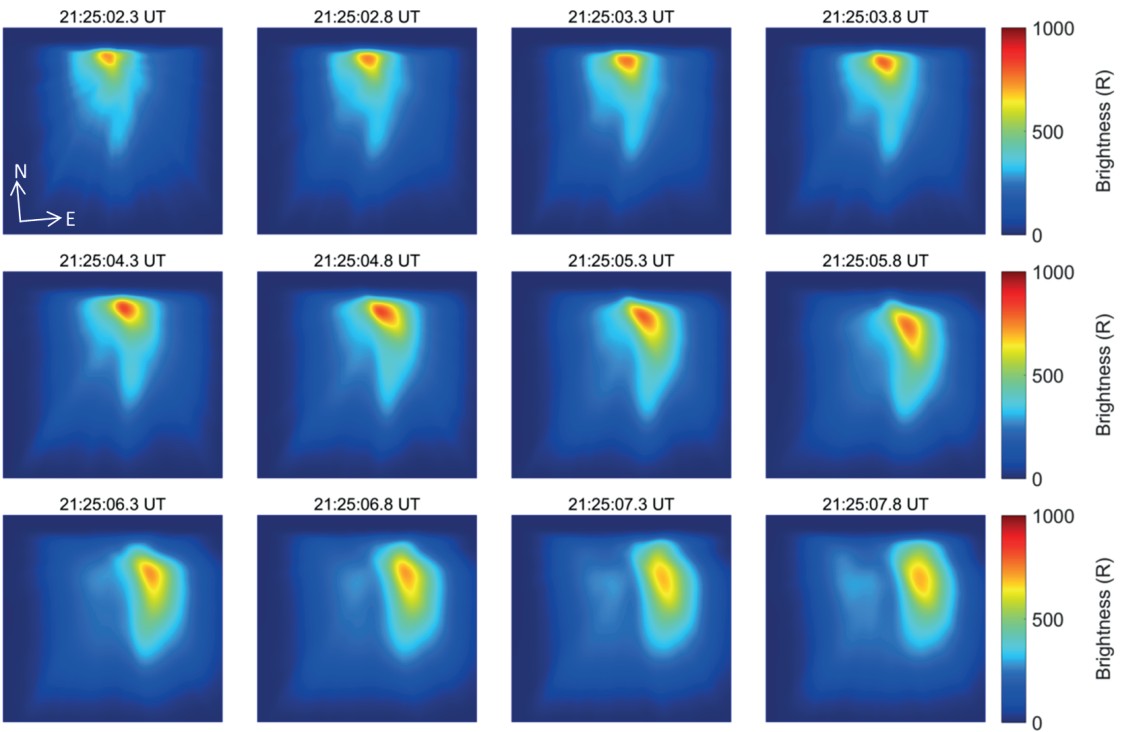

**Figure 6.** A sequence of modelled images of the 732.0 nm emission from the distribution of ions that convect away from their source region.





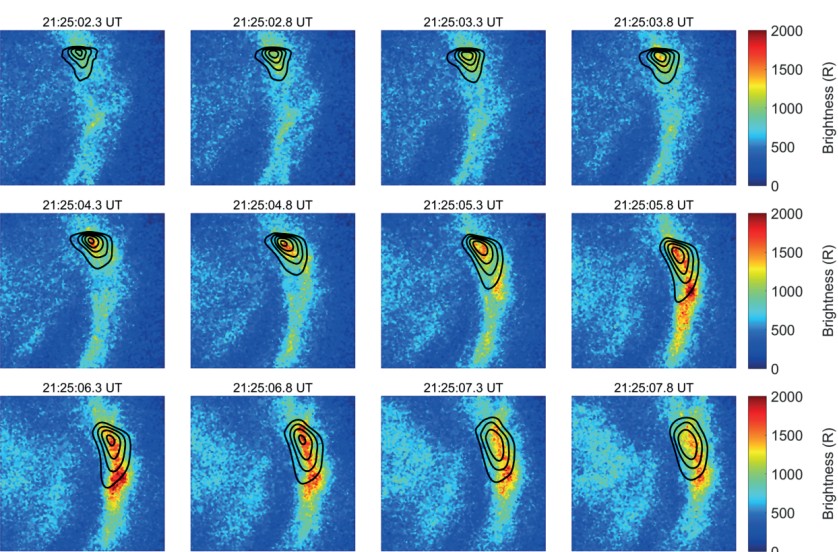

**Figure 7.** A sequence of observed images of the 732.0 nm emission. The black lines are model brightness contours from 400 R to 800 R at 100 R intervals.



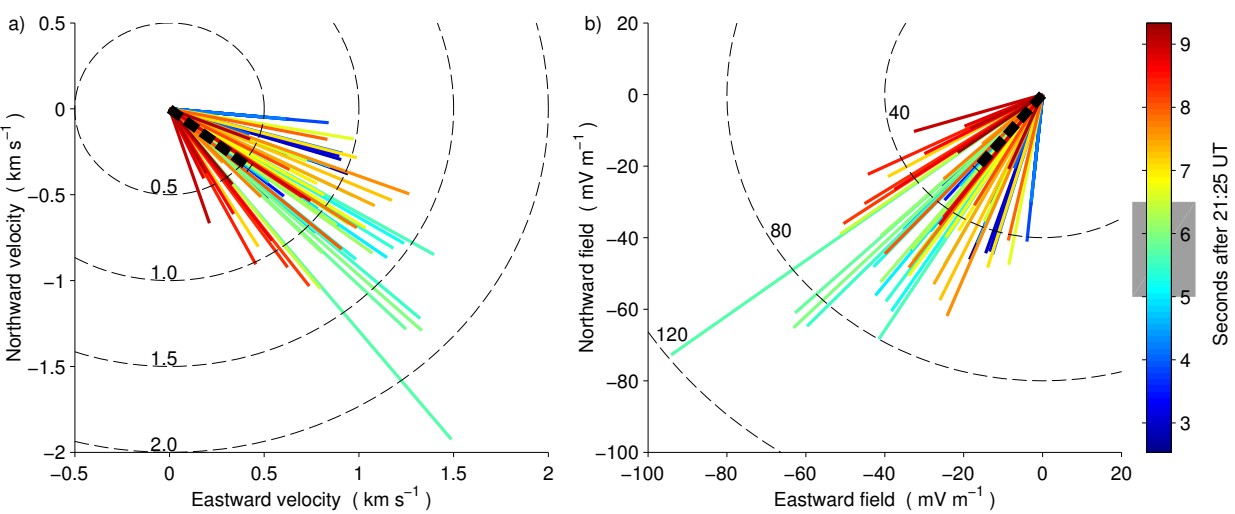

**Figure 8.** Magnitudes and directions of (a) the best-fit plasma velocities and (b) the inferred electric fields. The dashed circles are magnitude contours. The color of each line indicates the time of the velocity or field and the shaded region of the colorbar indicates when the arc brightened. The thick dot-dashed lines indicate the SuperDARN estimate of the average plasma velocity and electric field.



**Figure 9.** Merged plasma velocities over northern Scandinavia obtained between 21:24 and 21:26 UT on 9 November 2006. The grey line approximates the coastline of Norway, and the dotted lines are magnetic latitude and longitude. The red dots are at the position of the measured velocities and the direction of the velocity is given by the direction of the line from the dot. The length and colour of the lines indicate the magnitude of the plasma velocities. The black square is the approximate size and location of the ASK field of view at 200 km altitude. The SuperDARN data used in this figure were processed using RST version 2.11.

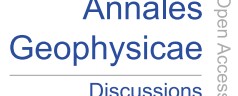
**Figure 10.** The large scale pattern of flow vectors between 21:24 and 21:26 UT on 9 November 2006. The position of the ASK instrument is marked.



**Figure 11.** Observed (left column) and modelled (right column) images of the auroral brightness $I_{673.0}$ (top row) and $I_{777.4}$) (bottom row) at 21:25:06 UT. The black lines on the modelled images are intensity contours of the observed images at 4 kR ($I_{673.0}$) and 2 kR ($I_{777.4}$). The white asterisks indicate the position of the magnetic zenith.





**Table 1.** Merged velocities from the Hankasalmi and Pykkvibaer radars between 21:24 UT and 21:26 UT on 9 November 2006 with coordinates, including those of ASK at 200 km.

| Velocity (ms$^{-1}$) | Azimuth angle | Magnetic Latitude | Magnetic Longitude |
|:---:|:---:|:---:|:---:|
| 621 | 126.8 | 67.5 | 100.4 |
| 622 | 137.8 | 67.5 | 103.0 |
| 430 | 154.5 | 66.5 | 101.3 |
| 639 | 141.0 | 66.5 | 103.8 |
| ASK | | 66.7 | 102.8 |