# Peer review of "Horizontal electric fields from flow of auroral $O^+(^2P)$ ions at sub-second temporal resolution"

_Annales Geophysicae, 2020_

## Referee Comment (RC1) · Anonymous Referee #1 · 13 Apr 2020

**General comments**

This work continues the efforts in the community of understanding the link between dynamic auroral features and the electric fields in the ionosphere connected to the electrodynamics of ionosphere-magnetosphere system. The work combines new techniques with existing techniques to come up with a new method of estimating ionospheric horizontal electric fields at high temporal resolution, associated with dynamic auroral features. The electric fields are inferred from plasma flow velocity which is got through a combination of ground based optical observations (ASK) at high spatial (100m) and temporal (0.05s), and modelling. Usually, these high temporal variations of the electric field (plasma flow velocities) associated with the dynamic auroral features are a challenge to capture. This paper precisely presents steps to derive the

plasma velocities incorporating cross validation with observations at different stages. In addition to the high resolution, by use of optical emissions observations at three wavelengths, it is possible to separate the brightening due motions of the source are from motion of the plasma. This has been a challenge in earlier work. Throughout the work presented, different steps have been taken to minimize the uncertainties, one of which is the correction of the position of the magnetic zenith.

Generally, the proof of concept has been precisely presented and supported by the large degree of agreement with observations for the case study used. Possible suggestions for improvement of method are also well presented. The work presented in this article is well written and important to the community. I have a few suggestions and comments.

**Specific comments**

For the title, may be add the word 'temporal' before 'resolution'

In lines 350-352 and 406-407 and Figure 8, It is stated that the superDARN velocity is representative of the background velocity based on the close agreement in magnitude and direction with average for period before and after brightness enhancement (i.e., outside shaded period on color bar in figure 8). However, this is true for direction but not clear for magnitude. **Suggestion:** Add an extra line in figure 8 or separate figure with just the black dashed line and a line showing the average for the period before and after the brightness enhancement (period outside the shaded period on color bar of figure 8). The close agreement with the background flow will be clearer to see.

**Technical corrections**

Line 2: Replace the word 'beside' with another word like 'associated with'
Line 30: Missing reference

Figure 1: Add a vertical axis label for panels d-f
Figure 9: Mention what numbers 1-4 mark in the figure caption.

Figure 10: Mention that the orange circle marks the ASK field of view

[Figure]

---

## Referee Comment (RC2) · Anonymous Referee #2 · 4 May 2020

Summary: In this article a method for estimating plasma drift around active aurora is presented using multi-spectral imaging and modelling. This builds on previous work on inferring electric fields in the ionosphere using imaging of the aurora. Two significant advances are made here. One is presenting several important considerations and improvements–such as being able to actually do this during the brightening of the arc which is critical. The other is the thoughtful charting of the necessary future steps. The current steps are clearly explained, including possible errors and future improvements and the conclusions are consistent with the analysis. Moreover, it is easy to understand why the the high spatial and temporal resolution is paramount in understanding auroral arc formation and why we are still not there. This work makes an essential contribution to that path and should be published with the most minor of adjustments.

• It is important to include is a statement explaining the motivation to invest in this method. It is hinted at but not explicitly stated (one might ask why not just use radar, having missed the point). Explicitly stating it in the abstract and conclusions would be sufficient to clarify the potential impact of this work.

Very Minor:

1) line 19: Please quantify or qualify "close".
2) line 30: Missing reference. Perhaps Clayton'18?
3) line 109: Year missing from last reference.
4) line 65: Please add a short explanation as to why that was not possible in order to make a better connection to your next point.
5) line 66: It would be good to add the specifics of the emissions as relevant in at least (1), (4), (5) for easier reference, particularly for the un-initiated.
6) line 148: Please check for consistency against lines 90, 102, 103. Maybe further clarification is needed in one or all of those places.
7) line 150: Are these not W-E keograms? Is there another reason why they are called stack plots and the term keogram is not avoided (is it because we usually see N-S and E-W)? Either way is fine of course.
8) line 214: Extra parentheses and some step missing?
8) line 278: Should this have been referenced in line 126?

---

## Author Comment (AC1) · 27 May 2020

We thank the referee for the very positive and helpful review. Our responses to the comments are below.

General comments

This work continues the efforts in the community of understanding the link between dynamic auroral features and the electric fields in the ionosphere connected to the electrodynamics of ionosphere-magnetosphere system. The work combines new techniques with existing techniques to come up with a new method of estimating ionospheric horizontal electric fields at high temporal resolution, associated with dynamic auroral features. The electric fields are inferred from plasma flow velocity which is

got through a combination of ground based optical observations (ASK) at high spatial (100m) and temporal (0.05s), and modelling. Usually, these high temporal variations of the electric field (plasma flow velocities) associated with the dynamic auroral features are a challenge to capture. This paper precisely presents steps to derive the plasma velocities incorporating cross validation with observations at different stages. In addition to the high resolution, by use of optical emissions observations at three wavelengths, it is possible to separate the brightening due motions of the source are from motion of the plasma. This has been a challenge in earlier work. Throughout the work presented, different steps have been taken to minimize the uncertainties, one of which is the correction of the position of the magnetic zenith.

Generally, the proof of concept has been precisely presented and supported by the large degree of agreement with observations for the case study used. Possible suggestions for improvement of method are also well presented. The work presented in this article is well written and important to the community. I have a few suggestions and comments.

Specific comments

For the title, may be add the word 'temporal' before 'resolution'

– This has been added

In lines 350-352 and 406-407 and Figure 8, It is stated that the superDARN velocity is representative of the background velocity based on the close agreement in magnitude and direction with average for period before and after brightness enhancement (i.e., outside shaded period on color bar in figure 8). However, this is true for direction but not clear for magnitude. Suggestion: Add an extra line in figure 8 or separate figure with just the black dashed line and a line showing the average for the period before and after the brightness enhancement (period outside the shaded period on color bar of figure 8). The close agreement with the background flow will be clearer to see.

[Figure]

– Thankyou for pointing out the need for a clearer emphasis of this comparison. On considering the various options, we decided that drawing one average value from the optical method to compare with one average from SuperDARN would involve several assumptions and approximations (as commented in the Discussion at lines 353-). Since the general comparison is valid within the constraints mentioned, we have added the actual numbers to be compared in the Discussion (line 360) as follows:

The optically derived velocities vary between 0.4 and 1.2 km s-1 in the few seconds either side of the arc brightening, compared with the average value of 0.6 km s-1 from SuperDARN.

Technical corrections

Line 2: Replace the word 'beside' with another word like 'associated with'

–To keep the notion of the proximity of the measured electric field to the arc we have written 'in the region close to (km scale)'. Note that Ref 2 required clarification of 'close' at start of introduction, which we prefer to change to 'in the region surrounding' as the distances there refer to various scales (as in other work).

Line 30: Missing reference – Latex error fixed

Figure 1: Add a vertical axis label for panels d-f – The label 'pixels' has been added to all panels.

Figure 9: Mention what numbers 1-4 mark in the figure caption. – Added to caption: The vectors labelled 1–4 are those closest to ASK as listed in Table 1.

Figure 10: Mention that the orange circle marks the ASK field of view – The caption has been changed to include the information that the orange circle marks the position of ASK (note: not the size of the field of view).

---

## Author Comment (AC2) · 27 May 2020

We thank the referee for the very positive and helpful review. Our response to comments are below, preceded with –.

Summary: In this article a method for estimating plasma drift around active aurora is presented using multispectral imaging and modelling. This builds on previous work on inferring electric fields in the ionosphere using imaging of the aurora. Two significant advances are made here. One is presenting several important considerations and improvements–such as being able to actually do this during the brightening of the arc which is critical. The other is the thoughtful charting of the necessary future steps. The current steps are clearly explained, including possible errors and future improvements

and the conclusions are consistent with the analysis. Moreover, it is easy to understand why the high spatial and temporal resolution is paramount in understanding auroral arc formation and why we are still not there. This work makes an essential contribution to that path and should be published with the most minor of adjustments.

$_I tis important to include is a statement explaining the motivation to invest in this method. It is hinted at but not explicitly stated (on$

– Given the importance of this point we have added the following sentences within the text.

In Abstract: They exhibit order-of-magnitude changes on temporal and spatial scales of seconds and kilometres which are not easy to measure; knowing their true magnitude and temporal variability is important for a theoretical understanding of auroral processes.

In Conclusions: Such high temporal resolution estimates of electric fields are a fundamental building block for the theory of auroral currents.

Also we have added to the Introduction (now line 49) the following sentence: These results demonstrated the need for a new method to estimate electric fields, and were key to the development of the method that is described here using high temporal resolution optical measurements.

The following sentence is already at the end of the Introduction (now line 64): The instrument used is the Auroral Structure and Kinetics (ASK) instrument, which was designed for the purpose of measuring plasma flows in a small 3.1×3.1 field of view around the magnetic zenith.

Very Minor:

line 19: Please quantify or qualify "close".

– 'of km scale' has been included in the abstract (responding to Ref 1) and the wording changed here to 'in the region surrounding' as this paragraph discusses several

different scales used by other methods.

line 30: Missing reference. Perhaps Clayton'18? – Latex error fixed.

line 109: Year missing from last reference. – Latex error fixed.

line 65: Please add a short explanation as to why that was not possible in order to make a better connection to your next point.

– Additional words (now line 69) are: They inferred electric fields of a few tens of $mV\,m^{-1}$ as an auroral event subsided. However, this method is limited by the fact that tracking is not possible during the main brightening because the motions of the source and the plasma cannot be separated without solving the continuity equation for the ions. The present method, referred to as the "flow model", overcomes the limitations of the above study through the following steps.

line 66: It would be good to add the specifics of the emissions as relevant in at least (1), (4), (5) for easier reference, particularly for the un-initiated.

– Emissions have been added in each case.

line 148: Please check for consistency against lines 90, 102, 103. Maybe further clarification is needed in one or all of those places.

– We have added to the words at line 97 (original line 90) to make clear that the brightness of the N2 emission does not vary with energy. The 777.4 nm emission results from both high and low energies through different processes, and with different sensitivities to energy, which is written at line108, and consistent with line 155 as follows:.

155 (148 original) ..the 673.0 nm (N2) and 777.4 nm (O) images, which measure the presence of high energy precipitation,

97 (90 original). . .and their brightness exhibits little dependence on the energy of the precipitating electrons.

108 (102 original) . . .the excitative process to be more sensitive to low energy pre­cipitation and the dissociative process to be more sensitive to high energy (> 1 keV) precipitation. This energy dependence results in emission from all precipitation ener­gies, but it is more responsive to low energy precipitation than the 673.0 nm emission.

line 150: Are these not W-E keograms? Is there another reason why they are called stack plots and the term keogram is not avoided (is it because we usually see N-S and E-W)? Either way is fine of course.

– Yes they are W-E keograms so have added for clarity.

line 214: Extra parentheses and some step missing?

– Thankyou for noticing two stray brackets

line 278: Should this have been referenced in line 126?

– No, this reference is not relevant to height integrating the emission profiles in the 1D model, which is a much simpler procedure.